# Neighbourhood-level socio-demographic characteristics and risk of COVID-19 incidence and mortality in Ontario, Canada: A population-based study

Trevor van Ingen[1], Kevin A. Brown[1,2,3], Sarah A. Buchan[1,2,3], Samantha Akingbola[1], Nick Daneman[1,2,4,5,6], Christine M. Warren[1], Brendan T. Smith[1,3]*

1 Public Health Ontario, Toronto, Ontario, Canada, 2 ICES, Toronto, Ontario, Canada, 3 Division of Epidemiology, Dalla Lana School of Public Health, University of Toronto, Toronto, Ontario, Canada, 4 Sunnybrook Research Institute, Sunnybrook Health Sciences Centre, Toronto, Ontario, Canada, 5 Division of Infectious Diseases, Sunnybrook Health Sciences Centre, Toronto, Ontario, Canada, 6 Department of Medicine, University of Toronto, Toronto, Ontario, Canada

* brendan.smith@oahpp.ca

**Data Availability Statement:** Public Health Ontario (PHO) cannot disclose the underlying data. Doing

## Abstract

### Objectives

We aimed to estimate associations between COVID-19 incidence and mortality with neighbourhood-level immigration, race, housing, and socio-economic characteristics.

### Methods

We conducted a population-based study of 28,808 COVID-19 cases in the provincial reportable infectious disease surveillance systems (Public Health Case and Contact Management System) which includes all known COVID-19 infections and deaths from Ontario, Canada reported between January 23, 2020 and July 28, 2020. Residents of congregate settings, Indigenous communities living on reserves or small neighbourhoods with populations <1,000 were excluded. Comparing neighbourhoods in the 90th to the 10th percentiles of socio-demographic characteristics, we estimated the associations between 18 neighbourhood-level measures of immigration, race, housing and socio-economic characteristics and COVID-19 incidence and mortality using Poisson generalized linear mixed models.

### Results

Neighbourhoods with the highest proportion of immigrants (relative risk (RR): 4.0, 95% CI:3.5–4.5) and visible minority residents (RR: 3.3, 95%CI:2.9–3.7) showed the strongest association with COVID-19 incidence in adjusted models. Among individual race groups, COVID-19 incidence was highest among neighbourhoods with the high proportions of Black (RR: 2.4, 95%CI:2.2–2.6), South Asian (RR: 1.9, 95%CI:1.8–2.1), Latin American (RR: 1.8, 95%CI:1.6–2.0) and Middle Eastern (RR: 1.2, 95%CI:1.1–1.3) residents. Neighbourhoods with the highest average household size (RR: 1.9, 95%CI:1.7–2.1), proportion of multigenerational families (RR: 1.8, 95%CI:1.7–2.0) and unsuitably crowded housing (RR: 2.1, 95%

so would compromise individual privacy contrary to PHO's ethical and legal obligations. Restricted access to the data may be available under conditions prescribed by the Ontario Personal Health Information Protection Act, 2004, the Ontario Freedom of Information and Protection of Privacy Act, the Tri-Council Policy Statement: Ethical Conduct for Research Involving Humans (TCPS 2 (2018)), and PHO privacy and ethics policies. Data are available for researchers who meet PHO's criteria for access to confidential data. Information about PHO's data access request process is available on-line at https://www. publichealthontario.ca/en/data-and-analysis/using-data/data-requests.

**Funding:** The author(s) received no specific funding for this work.

**Competing interests:** The authors have declared that no competing interests exist.

CI:2.0–2.3) were associated with COVID-19 incidence. Neighbourhoods with the highest proportion of residents with less than high school education (RR: 1.6, 95%CI:1.4–1.8), low income (RR: 1.4, 95%CI:1.2–1.5) and unaffordable housing (RR: 1.6, 95%CI:1.4–1.8) were associated with COVID-19 incidence. Similar inequities were observed across neighbourhood-level sociodemographic characteristics and COVID-19 mortality.

## Conclusions

Neighbourhood-level inequities in COVID-19 incidence and mortality were observed in Ontario, with excess burden experienced in neighbourhoods with a higher proportion of immigrants, racialized populations, large households and low socio-economic status.

## Introduction

Cumulating international evidence has documented disproportionately high rates of COVID-19 cases and mortality experienced by racialized and low income populations. For example, early in the pandemic a United Kingdom (UK) study found that Black adults have four times higher odds of COVID-19 mortality than White adults, with South Asian and mixed ethnicity individuals also having significantly higher odds of mortality [1]. In the United States (US), nearly 70% of early COVID-19 deaths in Chicago, Illinois occurred among Black individuals despite comprising only 30% of the population, with similar patterns observed in other heavily affected areas during the emergence of COVID-19 in the US [2]. During the early period of the pandemic, there was a dearth of US and Canadian COVID-19 surveillance data disaggregated according to important equity stratifiers, such as race and socio-economic status, and when available, was often incomplete [3, 4]. Better understanding the social distribution of COVID-19, particularly within local jurisdictions where COVID-19 trends differ, is critical towards informing the design of equitable policy and intervention strategies to reduce the burden of COVID-19 as well as planning for future pandemics.

Area-level measures of socio-economic status are commonly used to better understand inequities in COVID-19 incidence and mortality rates when individual-level measures are not available. Social epidemiological research has highlighted the role of neighbourhood characteristics in affecting health and contributing to socioeconomic and racial inequities in health [5]. Ecological studies using area-level measures have revealed widespread evidence of social inequities in COVID-19 incidence and mortality rates associated with race, poverty, and residential crowding [6–11]. While these trends have been consistently identified, the specific populations at risk and the strength of association vary between jurisdictions. Evidence is needed to understand the extent to which COVID-19 outcomes and risk factors vary according to neighborhood-level race and socio-economic status in the context of Ontario.

In Ontario–Canada's largest province by population–inequities in the burden of COVID-19 among both ethno-racially diverse and low socio-economic neighbourhoods have been identified [12–14]. While these findings confirm that socio-demographic factors influence COVID-19 risk in Ontario, the dichotomization of race based on neighbourhood-level 'diversity' and socio-economic categories based on neighbourhood-level 'material deprivation' in these reports impedes the ability to disaggregate findings by granular race and socio-economic categories. Previous Ontario studies have also not explored neighbourhood-level household characteristics, although overcrowding and multigenerational households have been suggested by local stakeholders as a cause of the higher burden faced by racialized communities in

Ontario [15]. Given that households are known to be an important source of COVID-19 transmission, and large households and apartment dwellers have been shown to have higher rates of COVID-19 mortality in Ontario, studying these associations may allow stakeholders to better understand and address COVID-19 inequities [16, 17]. Therefore, the objective of this study was to estimate the associations between neighbourhood socio-demographic characteristics and neighbourhood-level incidence and mortality of COVID-19 in Ontario during the first wave of the pandemic. Our primary objective was to estimate COVID-19 incidence and mortality rates across neighbourhood-level proportions of granular categories of immigration, race, housing, and socio-economic characteristics.

## Material and methods

### Study design and population

We conducted a population-based surveillance cohort study using data extracted from provincial and local reportable infectious disease surveillance systems, collectively known as the Public Health Case and Contact Management System (CCM) which include all known COVID-19 infections and deaths from Ontario, Canada reported between January 23, 2020 and July 28, 2020, the most recent data available at the time of the study.

The study population included all cases who met the provincial case definition for COVID-19 (i.e., positive nucleic acid amplification test). Due to the incomplete enumeration of Indigenous communities living on reserves in the Canadian Census (from which exposure and denominator data are derived), and the exclusion of people living in institutions and congregate living settings from the long-form census, these populations were excluded from this study [18, 19]. Residents of institutional and congregate living settings were identified and removed from our sample if they were flagged as such in CCM, or their residential address was matched to a comprehensive list of addresses of known institutions and congregate settings using a natural language processing algorithm. These settings and addresses include long-term care facilities, retirement homes, shelters, correction or detention centres, hotels and motels, group homes, hospitals, and on-site accommodations for farm workers.

### Covariables

Neighbourhood socio-demographic characteristics were derived from the 2016 Census of Population using Statistics Canada Aggregate Dissemination Areas (ADA) as the measure for neighbourhood. Twenty neighbourhoods with small populations (<1,000 people) were excluded due to stability concerns. On average, neighbourhood populations ranged between 5,000 to 15,000 people. Case records from CCM were assigned to a neighbourhood based on postal code of residence using Statistics Canada Postal Code Conversion File (PCCF) plus version 7C (i.e., November 2019 postal codes). A full description of the eighteen neighbourhood-level socio-demographic measures included in this study are available in S1 Table. These include: 1) eight measures of the proportion of immigration and race (immigrants, recent immigrants, visible minority (non-white and non-Indigenous population), Black, East/Southeast Asian, Latin American, Middle Eastern, and South Asian); 2) six measures of housing characteristics (average household size, proportion multigenerational families, proportion unsuitably crowded housing, proportion of dwellings in apartments in flat/duplex, proportion of dwellings in low-rise apartments, and proportion of dwellings in high-rise apartments); and 3) four socio-economic status measures (labour force participation, proportion without a high school diploma (age 25–64 years), proportion low income, and proportion unaffordable housing). Four categories of urban/rural geographic stratification (large urban centre, medium/small urban centre, rural, and remote) were determined by grouping neighbourhoods based

on community size, population density, and level of integration with a census metropolitan area or census agglomeration [20].

Age group (youth (<15 years old), working age (15–64 years old), and older adults (≥65 years old)) and sex (male, female) were extracted from CCM and included in the models as individual-level categorical variables.

## Outcomes

Cumulative (until July 28, 2020) incidence and mortality rates were calculated using the 2016 census population denominators, as more recent population projections were not available at the neighbourhood-level. Postal code of residence, case status, and outcome status were extracted from CCM. In CCM, COVID-19 deaths are defined as deaths resulting from a clinically compatible illness in a confirmed COVID-19 case, unless there is a clear alternative cause of death that cannot be related to COVID-19 (i.e., trauma).

The data used for the purposes of this project includes routinely collected COVID-19 case data. An authorized information custodian from Public Health Ontario anonymized the data before sharing it with the project team. Accordingly, individual consent was not required for the secondary use of non-identifiable information (TCPS 2 2018, Article 5.5B). This study received ethics clearance from Public Health Ontario's Research Ethics Board (File number: 2020–036.01).

## Statistical analysis

For each socio-demographic characteristic, the median percent and interdecile range of neighbourhood composition and the crude COVID-19 incidence and mortality rates per 100,000 population in the lowest and highest deciles were estimated. Further, associations between neighbourhood-level characteristics and neighbourhood-level counts of COVID-19 cases and deaths were estimated by fitting a series of Poisson generalized linear mixed models with random effects for neighbourhood, offset for neighbourhood population. Models were assessed for zero-inflation by comparing the observed number of zeroes with model predicted number of zeroes for all models. No models were found to be underfitting zeroes. Any overdispersion present in outcomes is accounted for by the use of random effects in all models [21]. Separate crude bivariable models were used to estimate associations between each neighbourhood measure of immigration, race, housing, and socio-economic status and COVID-19 incidence and mortality. Subsequently, each of these models were adjusted for individual-level age-group (<15, 15–64, and 65+ years) and sex (male/female), and neighbourhood-level urban/rural geography. To account for uneven distribution of socio-economic characteristics across neighbourhoods, all model estimates were standardized to show relative risks and 95% confidence intervals of COVID-19 incidence and mortality rates between the 10th (p10) and 90th (p90) percentile of each neighbourhood socio-demographic characteristic. All 95% confidence intervals were calculated using robust standard errors. The distribution of socio-demographic characteristics were plotted against COVID-19 incidence for each neighbourhood, along with solid lines representing the model-predicted estimates (derived using 'prediction' package in R) and dashed lines marking p10 and p90 for each predictor's distribution.

All analyses were conducted in R.

## Results

Between January 23 and July 28, 2020, 38,984 individuals with confirmed COVID-19 and 2,769 deaths were recorded in CCM in Ontario. Of those, 37,343 individuals (96%) had a valid postal code record that was successfully assigned to a neighbourhood. A further 8,822 (24%)

residents of congregate settings, and 59 (0.5%) Indigenous communities living on reserves or small neighbourhoods with populations <1,000 were excluded. In total, our study population included 28,808 COVID-19 cases and 683 COVID-19 deaths. Socio-demographic data were derived for 1,526 Ontario neighbourhoods.

## COVID-19 incidence and mortality across neighbourhoods

The distribution of COVID-19 incidence and mortality varied across neighbourhoods in Ontario (Fig 1). The top 10% highest incidence neighbourhoods accounted for 36% of the cases. The highest crude rate of COVID-19 incidence was 1,771 per 100,000. In nearly 70% of neighbourhoods, there were zero COVID-19 deaths and the top 10% highest mortality neighbourhoods accounted for 59% of all COVID-19 deaths. The highest crude neighbourhood COVID-19 mortality rate was 96 per 100,000.

The 18 neighbourhood-level measures we explored were described in terms of composition of the neighbourhoods (neighbourhood median and interdecile range), and incidence and mortality for neighbourhoods in the lowest and highest decile of that characteristic (Table 1). For most neighbourhood characteristics, the crude rates of COVID-19 incidence and mortality were higher among the highest compared to lowest decile, with the exception for neighbourhood low-rise apartments, and neighbourhood labour force participation. COVID-19 incidence was highest among the neighbourhoods in the highest decile of proportion Black residents, and mortality was highest among the neighbourhoods in the highest decile of unsuitably crowded housing.

## Distribution of neighbourhood-level characteristics and COVID-19 incidence

The crude rate of COVID-19 incidence per 100,000 were plotted as a function of neighbourhood-level immigration and race (Fig 2A), housing (Fig 2B), and socio-economic status (Fig 2C). Across 16 of the 18 predictors, as the proportion of the neighbourhood-level characteristic increases, so did the incidence of COVID-19. There was no association with proportion of

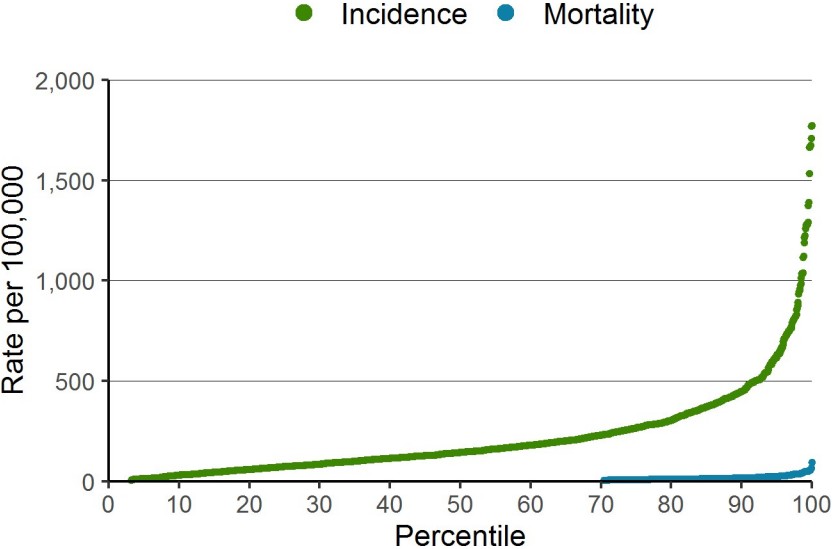

**Fig 1. Neighbourhood incidence and mortality rate per 100,000, ranked by neighbourhood percentile.**

**Table 1. Median neighbourhood socio-demographic characteristics, interdecile range, and incidence and mortality of COVID-19 in the lowest and highest deciles of a given neighbourhood characteristic.**

| Characteristic | Median (interdecile range) | Number of COVID-19 cases (incidence per 100,000) | | Number of COVID-19 deaths (mortality per 100,000) | |
|---|---|---|---|---|---|
| | | Lowest decile | Highest decile | Lowest decile | Highest decile |
| **Immigration and race** | | | | | |
| All immigrants (%) | 24.9 (6.9–59.0) | 683 (54.8) | 5,312 (392.1) | 14 (1.1) | 143 (10.6) |
| Recent immigrant (%) | 2.4 (0.3–8.5) | 818 (67.8) | 6,039 (434.0) | 24 (2.0) | 161 (11.6) |
| Visible Minority Status (%) | 20.2 (1.9–74.4) | 735 (59.9) | 6,754 (473.1) | 16 (1.3) | 141 (9.9) |
| Black (%) | 2.7 (0.4–11.8) | 849 (68.5) | 8,012 (583.7) | 18 (1.5) | 152 (11.1) |
| East/Southeast Asian (%) | 5.6 (0.6–22.7) | 912 (74.7) | 3,398 (243.0) | 20 (1.6) | 105 (7.5) |
| Latin American (%) | 1.1 (0.1–3.1) | 797 (65.5) | 7,111 (510.5) | 20 (1.6) | 157 (11.3) |
| Middle Eastern (%) | 1.4 (0.0–7.1) | 1,100 (66.4) | 3,705 (270.2) | 31 (1.9) | 93 (6.8) |
| South Asian (%) | 3.4 (0.3–25.2) | 975 (82.1) | 7,146 (506.9) | 28 (2.4) | 145 (10.3) |
| **Housing** | | | | | |
| Average household size (N) | 2.6 (2.1–3.5) | 2,902 (157.1) | 4,514 (413.6) | 89 (4.8) | 76 (7.0) |
| Multigenerational families (%) | 6.1 (2.5–16.3) | 1,700 (126.1) | 6,169 (460.6) | 43 (3.2) | 107 (8.0) |
| Unsuitably crowded housing (%) | 3.9 (1.7–13.8) | 935 (75.9) | 8,019 (566.8) | 24 (1.9) | 194 (13.7) |
| Apartment in duplex or flat (%) | 2.0 (0.3–8.3) | 2,539 (188.2) | 4,151 (333.3) | 65 (4.8) | 91 (7.3) |
| Low-rise apartment (%) | 6.0 (0.2–23.2) | 3,383 (252.9) | 2,965 (219.4) | 77 (5.8) | 72 (5.3) |
| High-rise apartment (%) | 3.8 (0.0–47.3) | 7,815 (170.9) | 5,257 (357.4) | 157 (3.4) | 166 (11.3) |
| **Socio-economic status** | | | | | |
| Labour force participation (%) | 64.9 (56.3–73.1) | 2,471 (190.2) | 2,547 (171.8) | 69 (5.3) | 42 (2.8) |
| Less than high school (%) | 17.0 (10.4–25.6) | 2,295 (155.8) | 4,721 (377.7) | 81 (5.5) | 98 (7.8) |
| Low income (%) | 7.7 (2.8–19.6) | 1,301 (97.8) | 5,333 (388.1) | 29 (2.2) | 144 (10.5) |
| Unaffordable housing (%) | 25.9 (15.9–39.2) | 1,436 (113) | 5,385 (363.3) | 32 (2.5) | 157 (10.6) |

low-rise apartments, and as neighbourhood proportion with less than high school education increases, COVID-19 decreases.

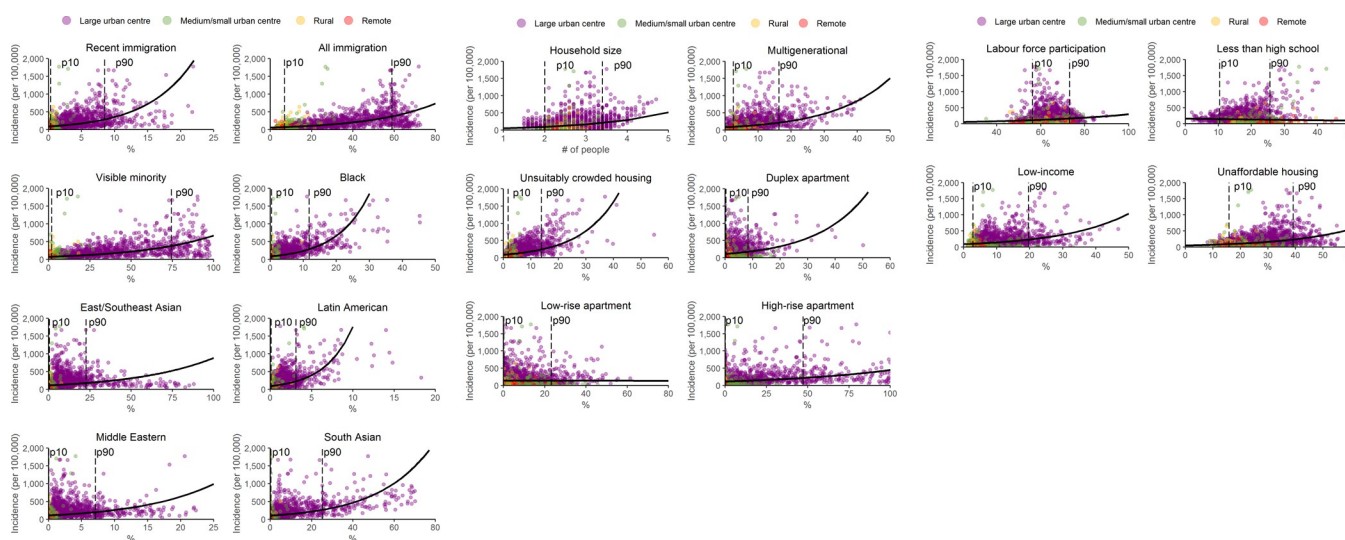

**Fig 2. a.** Neighbourhood-level incidence of COVID-19 by proportion of immigrant and race, with regression estimates and 10th (p10) and 90th (p90) percentiles. **b.** Neighbourhood-level incidence of COVID-19 by proportion of housing characteristics, with regression estimates and 10th (p10) and 90th (p90) percentiles. **c.** Neighbourhood-level incidence of COVID-19 by proportion of socio-economic status characteristics, with regression estimates and 10th (p10) and 90th (p90) percentiles.

## The associations between neighbourhood-level proportion of socio-demographic characteristics and COVID-19 incidence and mortality

Overall, crude and adjusted models estimated that neighbourhoods in the 90th percentile (p90) of socio-demographic characteristics were associated with higher rates of COVID-19 incidence and mortality compared to neighbourhoods with in the 10th percentile (p10) of socio-demographic characteristics for most unadjusted models (Table 2). Adjusting for age, sex, and urban/rural geographies reduced the strength of most associations.

The proportion of immigrants in a neighbourhood showed the strongest association with COVID-19, with an incidence relative risk of 4.0 (95% CI: 3.5–4.5) and mortality relative risk of 5.2 (95% CI: 4.1–6.7) in adjusted models. The proportion of visible minority residents showed stronger associations with incidence and mortality compared to the proportion of residents from individual race groups. Neighbourhood proportion of Black residents, followed by proportion of South Asian, Latin American, and to a lesser extent Middle Eastern residents showed the strongest associations of individual race groups. The proportion of East/Southeast Asian residents was associated with COVID-19 mortality not incidence in fully adjusting models.

Neighbourhoods with the highest average household size, proportion of multigenerational families and unsuitably crowded housing were associated with COVID-19 incidence and mortality in crude and adjusted models. Further, neighbourhoods with the highest proportion of high-rise apartments were associated with COVID-19 mortality, and to a lesser extent for

**Table 2. Relative risks of COVID-19 incidence and mortality between 10th and 90th percentile of each neighbourhood-level proportion of socio-demographic characteristics.**

| Characteristic | Incidence relative risk–p10 vs p90 (95% CI) | Adjusted* incidence relative risk–p10 vs p90 (95% CI) | Mortality relative risk–p10 vs p90 (95% CI) | Adjusted* mortality relative risk–p10 vs p90 (95% CI) |
|---|---|---|---|---|
| **Immigration and race** | | | | |
| Recent immigrant | 3.2 (2.9–3.6) | 2.1 (1.9–2.4) | 2.8 (2.4–3.3) | 2.5 (2.1–3.0) |
| All immigrants | 5.4 (4.8–6.0) | 4.0 (3.5–4.5) | 5.2 (4.1–6.7) | 5.2 (3.9–7.0) |
| Visible Minority Status | 5.0 (4.5–5.6) | 3.3 (2.9–3.7) | 3.7 (3.0–4.7) | 3.5 (2.7–4.5) |
| Black | 3.2 (2.9–3.5) | 2.4 (2.2–2.6) | 2.2 (1.9–2.4) | 2.0 (1.7–2.2) |
| East/Southeast Asian | 1.6 (1.4–1.7) | 1.0 (1.0–1.1) | 1.5 (1.3–1.7) | 1.2 (1.0–1.3) |
| Latin American | 2.4 (2.2–2.7) | 1.8 (1.6–2.0) | 1.9 (1.6–2.1) | 1.6 (1.4–1.8) |
| Middle Eastern | 1.9 (1.7–2.1) | 1.2 (1.1–1.3) | 1.6 (1.4–1.8) | 1.3 (1.1–1.5) |
| South Asian | 2.6 (2.4–2.8) | 1.9 (1.8–2.1) | 1.9 (1.7–2.2) | 1.8 (1.6–2.1) |
| **Housing** | | | | |
| Average household size | 2.2 (2.1–2.3) | 1.9 (1.7–2.1) | 1.5 (1.1–2.0) | 1.6 (1.3–2.0) |
| Multigenerational families | 2.2 (2.0–2.4) | 1.8 (1.7–2.0) | 1.8 (1.5–2.1) | 1.7 (1.4–2.0) |
| Unsuitably crowded housing | 2.4 (2.2–2.7) | 2.1 (2.0–2.3) | 2.5 (2.2–2.9) | 2.5 (2.1–2.9) |
| Apartment in duplex or flat | 1.5 (1.4–1.7) | 1.3 (1.2–1.3) | 1.4 (1.2–1.6) | 1.2 (1.0–1.4) |
| Low-rise apartment | 1.0 (0.9–1.1) | 0.9 (0.8–1.0) | 1.1 (0.9–1.4) | 1.0 (0.8–1.2) |
| High-rise apartment | 1.9 (1.8–2.1) | 1.2 (1.1–1.4) | 2.3 (1.9–2.7) | 1.7 (1.4–2.1) |
| **Socio-economic status** | | | | |
| Labour force participation | 1.5 (1.4–1.5) | 1.0 (0.9–1.1) | 0.7 (0.5–0.8) | 0.8 (0.6–1.0) |
| Less than high school education | 0.8 (0.8–0.9) | 1.6 (1.4–1.8) | 1.3 (1.2–1.6) | 2.1 (1.7–2.6) |
| Low income | 2.1 (1.9–2.3) | 1.4 (1.2–1.5) | 2.3 (1.9–2.7) | 1.8 (1.5–2.2) |
| Unaffordable housing | 2.6 (2.3–3.0) | 1.6 (1.4–1.8) | 3.1 (2.5–3.9) | 2.4 (1.9–3.0) |

* Adjusted for age-group, sex, and urban/rural stratifier

COVID-19 incidence and between neighbourhoods with the highest proportion of apartment in duplex or flat and COVID-19 incidence and mortality in adjusted models. The proportion of low-rise apartments was not associated with COVID-19 incidence or mortality.

Neighbourhoods with the highest proportion of residents with less than high school, low income and unaffordable housing was associated with COVID-19 incidence and mortality. Neighbourhoods with high labour force participation had lower rates of COVID-19 mortality compared to neighbourhoods with low labour force participation. In adjusted models the protective association of neighbourhoods with the highest compared to lowest proportion of less than high school education was inversed, indicating an association with increased COVID-19 incidence.

## Discussion

Linking COVID-19 surveillance data to neighbourhood-level characteristics from Ontario, Canada between January 23, 2020 and July 28, 2020, this study found higher COVID-19 incidence and mortality rates in neighbourhoods with a higher proportion of immigrants, racialized populations, large households and low socio-economic status. These findings highlight how neighbourhood-level conditions, which reflect social environments that are influenced by institutional and structural systems (e.g., policy) [22], act as key determinants of COVID-19 inequities.

People living in the most marginalized neighbourhoods are experiencing elevated rates of COVID-19 outcomes, both in Canada [12, 13, 23] and internationally [6, 7, 11, 24–26]. For example, US counties with the highest compared to the lowest proportion of populations of colour and poverty had 4.9 and 1.7 times higher rates of COVID-19 death [7]. Similarly in Canada, age-standardized COVID-19 mortality rates were two times higher in neighbourhoods with the highest (>25%) compared to lowest (<1%) proportion of visible minorities, although this varied by province with COVID-19 mortality rates 3.4 higher in Ontario [27]. Our study adds to this body of evidence by examining associations between specific socio-demographic characteristics and COVID-19 burden. We found neighbourhoods with a high proportion of immigrants had four times higher risk of COVID-19 infection and 5.2 times higher risk of death, followed by neighbourhoods with a high proportion of visible minority residents which had 3.3 times higher risk of COVID-19 incidence and 3.5 times higher risk of death. Further, the increased risk among neighbourhoods with high proportions of Black, South Asian, and Latin American populations found in our study are consistent with a a recent systematic-review and meta-analysis of 50 studies on individual-level ethnicity and COVID-19 from the UK and US [28].

Existing and persistent structural inequities put immigrant, racialized, and low-income communities at higher risk of COVID-19 exposure and infection. Housing conditions, especially household size and crowding, are an important predictor of COVID-19 transmission [7, 16, 29]. In Canada, 21.1% of racialized individuals live in unsuitably crowded households, nearly four times the rate of non-racialized individuals [30]. Additionally, immigrants are twice as likely to live in multigenerational households compared to non-immigrants [31]. In our study, neighbourhood-level housing characteristics were associated with increasing risk of COVID-19 incidence and mortality. Future research is required to examine the extent to which housing characteristics explain the disproportionate impact of COVID-19 on immigrant, racialized, and low-income communities.

Low socio-economic status further accounts for excess COVID-19 burden experienced by immigrant and racialized populations, impacting the ability to avoid COVID-19 infection at work. Evidence from the literature describes significant risks of COVID-19 associated with

occupations, especially those in precarious employment, where it is difficult to distance from others, there is increased risk of exposure to infections while at work, and employers are less likely to provide paid sick leave [32–35]. In Canada, immigrant and racialized individuals are not only more likely to live in poverty [36], but also overrepresented in risky essential occupations. For example, immigrants and racialized populations in Ontario are disproportionately employed in long-term care facilities as nurse aids, orderlies, and patient service associates [37]. Further, COVID-19 testing data from Ontario confirmed that a disproportionate number of immigrants diagnosed with COVID-19 were employed as health care workers [38]. Exploring the role that occupation and workplace settings has on contributing to inequities in COVID-19 transmission and negative outcomes in Ontario represents an important area of future study.

Other explanations for the stark inequities in COVID-19 outcomes found in this study may be rooted in systematic barriers faced by racialized and newcomer populations [39]. In Canada, racism and other structural determinants are an underlying cause of the overrepresentation of Black individuals having lower socio-economic status and inadequate access to a regular doctor [40, 41]. Racialized populations also experience pre-existing health inequities, such as higher rates of comorbidities, which could also contribute to greater vulnerability to severe COVID-19 outcomes [42, 43]. However, data from the first wave in Ontario showed that although incidence and mortality were higher among diverse neighbourhoods, the case-fatality ratio was lower, suggesting that greater number of infections, and not co-morbidities, was a driving cause for the increased rate of death [12]. Systemic racism has been observed in other Western jurisdictions in ways that can contribute to increased COVID-19 risk. In the US, racialized individuals are more likely to be incarcerated, which in turn increases the likelihood of infection [8], and residential segregation, caused in part by discriminatory mortgage lending practices in urban areas, may be a driving factor in residential crowding among Black Americans [44]. In the UK, non-White physicians comprise the vast majority of COVID-19 deaths among doctors and, compared to White physicians, are more likely to report being under pressure to attend to patients without receiving the necessary physical protection [45].

This study is subject to some limitations. First, the number of COVID-19 cases included in this study is an undercount of the true number of cases, and may be biased by changes in testing criteria during the study period. Testing criteria for SARS-CoV-2 during the study period shifted from initially being restricted to identifying cases in returning symptomatic travelers or individuals with direct exposure to a recent traveler, to being broadly expanded to include asymptomatic individuals in May 2020 [46]. These changing criteria may have contributed to observed differences in testing patterns between various socio-demographic populations. An Ontario study undertaken concurrently with the current study's period of observation found decreased odds of having been tested for COVID-19 (i.e. communities with higher percentages of lower income and visible minorities) and increased odds of having received a positive COVID-19 diagnosis (i.e., increase quintile of people per dwelling and with limited education attainment) in models adjusted for age, sex, underlying health conditions, previous health care, public health region, environment and area-based social determinants of health [47]. The resulting under detection suggest the associations between neighbourhood socio-demographic factors and COVID-19 incidence and mortality in our study are likely conservative. Exclusion of people living in institutions, congregate living settings, and Indigenous communities living on reserves also likely resulted in an underestimate of the number of cases disproportionately across socio-demographic indicators, however these groups were excluded due to poor representation in the Census. Additionally, our area-level findings should not be interpreted at the individual-level, as individual cases may not reflect the characteristics of the neighbourhoods they live in. Previous Canadian studies comparing individual and area-level measures have

shown that even with relatively poor agreement between measures, area-level measures may be describing important community-level effects that contribute to health inequities [48]. The collection of individual-level socio-demographic data in Ontario during subsequent phases of the pandemic will allow for future validation of our findings. Moreover, structural social and economic conditions shared by individuals provide an opportunity for decision-makers to create influential policies related to reducing health disparities [22]. Finally, results from our observational study do not allow for causal relationships to be assessed.

This study has several strengths. The ability to use postal code to link people with COVID-19 to census data at the neighbourhood-level provides greater accuracy than is possible with most publically available data on COVID-19. Additionally, we included a large number of socio-demographic characteristics, notably the ability to examine specific immigration, race, housing and socio-economic categories.

## Conclusion

Neighbourhood socio-demographic factors, including immigration, race, housing and socio-economic status are associated with COVID-19 incidence and mortality in Ontario. These results suggest that culturally safe approaches to engaging with immigrant, racialized and low socio-economic status communities are important public health strategies for reducing COVID-19 inequities. Future research on COVID-19 inequities should focus how the relationship between the socio-demographic factors examined in this study and COVID-19 are confounded by occupation and workplace characteristics.

## Supporting information

**S1 Table. Neighbourhood-level socio-demographic characteristic variable descriptions.** (PDF)

## Author Contributions

**Conceptualization:** Trevor van Ingen, Kevin A. Brown, Sarah A. Buchan, Nick Daneman, Brendan T. Smith.

**Formal analysis:** Trevor van Ingen, Kevin A. Brown.

**Methodology:** Trevor van Ingen, Kevin A. Brown, Sarah A. Buchan, Samantha Akingbola, Nick Daneman, Christine M. Warren, Brendan T. Smith.

**Writing – original draft:** Trevor van Ingen, Samantha Akingbola.

**Writing – review & editing:** Trevor van Ingen, Kevin A. Brown, Sarah A. Buchan, Samantha Akingbola, Nick Daneman, Christine M. Warren, Brendan T. Smith.

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
