## [Decision Letter · Decision Letter 0]

18 Jul 2022

PONE-D-22-16493Neighbourhood-level socio-demographic characteristics and risk of COVID-19 incidence and mortality in Ontario, Canada: a cross-sectional studyPLOS ONE

Dear Dr. Smith,

Thank you for submitting your manuscript to PLOS ONE. After careful consideration, we feel that it has merit but does not fully meet PLOS ONE’s publication criteria as it currently stands. Therefore, we invite you to submit a revised version of the manuscript that addresses the points raised during the review process.

We look forward to receiving your revised manuscript.

Kind regards,

Csaba Varga, DVM MSc PhD

Academic Editor

PLOS ONE

Journal Requirements:

2. In the ethics statement in the manuscript and in the online submission form, please provide additional information about the patient records/samples used in your retrospective study. Specifically, please ensure that you have discussed whether all data/samples were fully anonymized before you accessed them and/or whether the IRB or ethics committee waived the requirement for informed consent. If patients provided informed written consent to have data/samples from their medical records used in research, please include this information.

Reviewers' comments:

Reviewer's Responses to Questions

**Comments to the Author**

1. Is the manuscript technically sound, and do the data support the conclusions?

Reviewer #1: Partly

Reviewer #2: Yes

Reviewer #3: Yes

2. Has the statistical analysis been performed appropriately and rigorously? 

Reviewer #1: Yes

Reviewer #2: Yes

Reviewer #3: Yes

3. Have the authors made all data underlying the findings in their manuscript fully available?

Reviewer #1: No

Reviewer #2: Yes

Reviewer #3: No

4. Is the manuscript presented in an intelligible fashion and written in standard English?

Reviewer #1: Yes

Reviewer #2: Yes

Reviewer #3: Yes

5. Review Comments to the Author

Reviewer #1: Thank you for the opportunity to review this manuscript. The authors describe associations between neighbourhood-level social and demographic characteristics and COVID-19 incidence and mortality rates over a 7-month period at the beginning of the pandemic. This study provides a more granular understanding of the relationship between specific neighbourhood characteristics and COVID-19 outcomes and provides additional evidence for social inequities. The study benefits from access to complete COVID-19 infection and death information over the study period and utilization of multiple Census measures of immigration, race, housing, and socio-economic characteristics. The authors were importantly able to adjust for individual-level age and sex and neighbourhood-level urban/rural variables. Please find additional comments for the authors’ consideration to strengthen the manuscript.

General:

Cross-sectional study designs do not generally support the measurement of incidence or relative risk. Can the authors be more specific in their interpretation or provide an explanation for how the study design supports calculation of disease incidence and relative risk?

Abstract:

Lines 31-37: Please include major findings related to housing and socio-economic characteristics in addition to the findings you have presented for neighbourhood-level measures of immigration and race.

Materials and Methods:

Line 87: Indicate rationale for dates selected.

Line 104: Indicate Version 7c is for November 2019 postal codes.

Line 104: Provide a reference to Census variable definitions (e.g. What is meant by low income or unaffordable housing?)

Line 113: Indicate four categories and how these map onto binary urban/rural classification.

Line 115: Provide rationale for age cut-off values and modelling age as a categorical rather than continuous variable.

Line 132: Please confirm whether the authors checked and confirmed that over-dispersion was not an issue such that the Poisson distribution is appropriate.

Line 137: Indicate how 95% confidence intervals were calculated. Did you use robust standard errors?

Line 138: Provide rationale for p10/p90 comparison.

Results:

Where possible, please clarify whether you are referring to crude or adjusted rates.

Line 174-175: Specifically state what the solid black line and dotted lines represent in Figure 2. Include in the Methods section how the trend line was estimated.

Lines 195-196: Indicate the other characteristics that were also not associated with incidence and/or mortality, particularly after adjustment.

Lines 198-199: You are also controlling for age and sex as confounders, which are also likely playing an important role here.

Lines 211+: Include a summary of results for housing and socio-economic status as well.

Figures 2A-C – Include the 95% confidence interval

Figures 2A-C – Crop x-axes where there are no data points (Middle Eastern, less than high school are notable)

Figures 2A-C – Label all vertical lines (p10, p90)

Discussion:

Line 218: Not directly. Your study examines neighbourhood-level characteristics as determinants of inequities but does not examine specific structural barriers. Suggest rewording for accuracy.

Lines 238-241: If I understand how you constructed your models correctly, your study does not support this conclusion. Did you construct models that included immigration, race, and housing variables together? As I interpreted Table 2, you had separate models for 1) immigration and race, 2) housing, and 3) socio-economic status variables where these groupings were adjusted for age, sex, and urban/rural status. Please clarify.

Line 270: Please include any limitations associated with using a cross-sectional design.

Line 271: Were there changes in testing criteria during this time period? If so, please clarify what these changes were.

Line 275: I think this misrepresents the findings of Sundaram’s study as significant associations for testing and testing positive were indeed found in fully adjusted models, particularly for variables of importance to this study. I would suggest reconsidering the role of selection bias in your findings.

Line 280: I am not sure what you mean by the term ‘dilute’. Is there an alternate term that can be used?

Conclusion:

Line 296: Include immigration as a key neighbourhood characteristic.

Line 296: This is the first time you mention poverty and this may not be the most accurate term to use here – perhaps low SES or low income communities would be more appropriate to the study context.

Reviewer #2: This paper was an enjoyable read and a good example of how combing datasets can provide valuable insights for policy makers. The methods used to find associations between neighborhood-level sociodemographic measures and COVID-19 incidence and mortality are well articulated and based on sound statistical methods. The comments below are minor and aim to improve the clarity of the paper for the reader. I recommend publications with minor revision.

- Lines 33-37: In abstract, IRRs are difficult to interpret as referent and comparison groups are not clear (what do you mean by high proportion). Including details about the deciles would be helpful (comparing 10p - 90p).

- Line 46: Make it more clear that the South Asian finding was part of the same UK study and make it clear who they have higher odds of mortality compared to

- Line 64: Perhaps ‘context of Ontario’ is more appropriate than ‘Canadian context’, as you highlight the need for jurisdiction specific findings.

- Line 87: Can you provide a rationale for the dates chosen to define your study period?

- Lines 89-92: Do you have a citation to support the statement that the census has poor representation of those groups?

- Lines 104-112: Can you include (in appendix or refer to a published source) definitions used for the socio-demographic measures? Many are self explanatory, but some questions I have include what counts as recent immigration? What defines unsuitably crowded housing? What defines low income? What defines unaffordable housing?

- Line 113: Listing the four categories would be helpful.

- Line 141: Information on what if any model diagnostics or assessments of goodness of fit tests were done on the models is needed. Was there over-dispersion?

- Line 148: How many deaths were in the initial dataset before removing congregate settings? You state 24% of cases were in these settings, but knowing the proportion of deaths provides important context for the overall mortality findings.

- Line 149: How many Ontario neighborhoods were excluded from the analysis because they were too small? And can you speak to what the characteristics of these small neighborhoods compared to those included in this study? Could this introduce any bias to your results?

- Line 189: The term multivariable model causes some confusion as they are interchanged with ‘adjusted model’ and ‘fully adjusted model’ in the tables and throughout the paper. I find ‘adjusted model’ to be the most clear in this case.

- Line 196: I understand what is being referred to when saying the adjusting the model ‘reversed’ the protective association but I do not know if it is an accurate term to use. Perhaps ‘inversed’ would be better.

- Line 217: I believe your findings do show that neighbourhood level factors are key determinants of COVID-19 inequities, but I am not clear how your findings show ‘how structural barriers are acting as key determinants of COVID-19 inequities”. I would suggest narrowing the conclusions, or elaborating more on how your findings do support that conclusion (ie. Can you provide an example of a specific structural barrier that acted as a determinant?)

- Lines 238-241: It sounds like this finding comes from an additional analysis that was not described in the methods and not shown in the results? I believe this analysis is of extreme interest. Being able to show that the socio-demographic factors are intertwined but still independently significant even when controlling for other socio-demographic factors is a major finding. It could also provide important evidence for policy makers (just addressing housing will not erase inequalities). I understand it may be difficult to include given word count limitations, but this analysis would be highly interesting.

- Lines 270-275: Glad to see you addressed the risk of differential COVID testing biasing the results, as this was a concern of mine reading the paper. You did a very good job addressing this concern.

- Lines 294-295: The claim that socio-demographic factors explain much of the neighbourhood-level variability in COVID-19 needs further evidence. Reading this claim, it sounds like if you were to build a multivariable model with all the socio-demographic measures as explanatory variables, your R-Squared (or pseudo R-Squared) would be greater that 50%. Did you find this? If so, that is great but please elaborate in the results.

- Lines 295-296: The conclusion that “culturally safe approaches to engaging with racialized communities and communities living in poverty, are important public health strategies for reducing COVID-19 inequities” is not fully substantiated by your findings. Your findings have identified the problem (neighborhood-level measures are associated with COVID inequalities) but I believe cannot go as far to suggest what will solve the problem. I believe your research will provide meaningful information to inform public health strategies.

Reviewer #3: This paper provides a unique look at COVID-19 data at a level not examined in many publications. This work is an important piece of the puzzle in understanding infectious disease mitigation in an outbreak scenario. Congratulations to the authors for producing this quality work

6. PLOS authors have the option to publish the peer review history of their article (what does this mean?). If published, this will include your full peer review and any attached files.

Reviewer #1: No

Reviewer #2: No

Reviewer #3: No

---

## [Author Response · Author response to Decision Letter 0]

15 Sep 2022

We have responded to all reviewer and editor comments in our attached "Response to reviewers document".

---

## [Decision Letter · Decision Letter 1]

10 Oct 2022

Neighbourhood-level socio-demographic characteristics and risk of COVID-19 incidence and mortality in Ontario, Canada: a population-based study

PONE-D-22-16493R1

Dear Dr. Brendan Smith,

We’re pleased to inform you that your manuscript has been judged scientifically suitable for publication and will be formally accepted for publication once it meets all outstanding technical requirements.

Kind regards,

Csaba Varga, DVM MSc PhD

Academic Editor

PLOS ONE

Additional Editor Comments (optional):

Reviewers' comments:

Reviewer's Responses to Questions

**Comments to the Author**

1. If the authors have adequately addressed your comments raised in a previous round of review and you feel that this manuscript is now acceptable for publication, you may indicate that here to bypass the “Comments to the Author” section, enter your conflict of interest statement in the “Confidential to Editor” section, and submit your "Accept" recommendation.

Reviewer #1: All comments have been addressed

Reviewer #2: All comments have been addressed

2. Is the manuscript technically sound, and do the data support the conclusions?

Reviewer #1: (No Response)

Reviewer #2: Yes

3. Has the statistical analysis been performed appropriately and rigorously? 

Reviewer #1: (No Response)

Reviewer #2: Yes

4. Have the authors made all data underlying the findings in their manuscript fully available?

Reviewer #1: (No Response)

Reviewer #2: Yes

5. Is the manuscript presented in an intelligible fashion and written in standard English?

Reviewer #1: (No Response)

Reviewer #2: Yes

6. Review Comments to the Author

Reviewer #1: (No Response)

Reviewer #2: (No Response)

7. PLOS authors have the option to publish the peer review history of their article (what does this mean?). If published, this will include your full peer review and any attached files.

Reviewer #1: No

Reviewer #2: No

---

## [Editor Report · Acceptance letter]

13 Oct 2022

PONE-D-22-16493R1 

Neighbourhood-level socio-demographic characteristics and risk of COVID-19 incidence and mortality in Ontario, Canada: a population-based study 

Dear Dr. Smith:

I'm pleased to inform you that your manuscript has been deemed suitable for publication in PLOS ONE. Congratulations! Your manuscript is now with our production department. 

Kind regards, 

on behalf of

Dr. Csaba Varga 

Academic Editor

PLOS ONE